# Açaí (*Euterpe oleracea* Mart.) Seed Extracts from Different Varieties: A Source of Proanthocyanidins and Eco-Friendly Corrosion Inhibition Activity

**DOI:** 10.3390/molecules26113433

**Published:** 2021-06-05

**Authors:** Gabriel Rocha Martins, Douglas Guedes, Urbano Luiz Marques de Paula, Maria do Socorro Padilha de Oliveira, Marcia Teresa Soares Lutterbach, Leila Yone Reznik, Eliana Flávia Camporese Sérvulo, Celuta Sales Alviano, Antonio Jorge Ribeiro da Silva, Daniela Sales Alviano

**Affiliations:** 1Instituto de Pesquisas de Produtos Naturais, Bloco H, Centro de Ciências da Saúde, Universidade Federal do Rio de Janeiro. Av. Carlos Chagas Filho, 373, Cidade Universitária, Rio de Janeiro 21941-902, Brazil; gabrielrmartins@gmail.com (G.R.M.); urbanomarques@outlook.com (U.L.M.d.P.); ajorge@ippn.ufrj.br (A.J.R.d.S.); 2Escola de Química, Bloco E, Centro de Tecnologia, Universidade Federal do Rio de Janeiro. Av. Athos da Silveira Ramos, 149, Cidade Universitária, Rio de Janeiro 21941-909, Brazil; douglas.guedes.ferreira@gmail.com (D.G.); lreznik@eq.ufrj.br (L.Y.R.); eliana@eq.ufrj.br (E.F.C.S.); 3Embrapa Amazônia Oriental–Trav. Dr. Enéas Pinheiro, s/n.–Belém, Pará 66095-100, Brazil; socorro-padilha.oliveira@embrapa.br; 4Instituto Nacional de Tecnologia, Divisão de Degradação e Corrosão. Av. Venezuela, 82, Rio de Janeiro 20081-312, Brazil; marcia.lutterbach@int.gov.br; 5Instituto de Microbiologia Paulo de Góes, Bloco I, Centro de Ciências da Saúde, Universidade Federal do Rio de Janeiro. Av. Carlos Chagas Filho, 373, Cidade Universitária, Rio de Janeiro 21941-902, Brazil; alviano@micro.ufrj.br

**Keywords:** proanthocyanidins, degree of polymerization, green corrosion inhibitor, white açaí, BRS-Pará, phenolic compounds, residue exploitation, mass spectrometry analysis, açaí seeds

## Abstract

*Euterpe oleracea* Mart. (Arecaceae) is an endogenous palm tree from the Amazon region. Its seeds correspond to 85% of the fruit’s weight, a primary solid residue generated from pulp production, the accumulation of which represents a potential source of pollution and environmental problems. As such, this work aimed to quantify and determine the phytochemical composition of *E. oleracea* Mart. seeds from purple, white, and BRS-Pará açaí varieties using established analytical methods and also to evaluate it as an eco-friendly corrosion inhibitor. The proanthocyanidin quantification (*n*-butanol/hydrochloric acid assay) between varieties was 6.4–22.4 (*w*/*w*)/dry matter. Extract characterization showed that all varieties are composed of B-type procyanidin with a high mean degree of polymerization (mDP ≥ 10) by different analytical methodologies to ensure the results. The purple açaí extract, which presented 22.4% (*w*/*w*) proanthocyanidins/dry matter, was tested against corrosion of carbon steel AISI 1020 in neutral pH. The crude extract (1.0 g/L) was effective in controlling corrosion on the metal surface for 24 h. Our results demonstrated that the extracts rich in polymeric procyanidins obtained from industrial açaí waste could be used to inhibit carbon steel AISI 1020 in neutral pH as an abundant, inexpensive, and green source of corrosion inhibitor.

## 1. Introduction

*Euterpe oleracea* Mart., an endogenous palm tree of the Amazonian biome that grows in flooded areas, produces a dark purple fruit called açaí [1]. Its edible pulp is rich in antioxidants and nutrients to the most impoverished local populations [2] and now commercialized as a functional food [3]. Açaí has high nutritional value due to the predominance of fatty acids and amino acids. However, the antioxidant capacities of high contents of anthocyanins (cyanidin 3-glucoside and cyanidin 3-rutinoside) and other flavonoids are associated with açaí’s health benefits against several diseases related to oxidative stress [4].

Traditionally, açaí is a purple globose fruit. However, it can be found in a green to yellow color with a crème mesocarp, commonly referred to as “white açaí” [3,5]. Additionally, Embrapa Eastern Amazonia (Belém, Pará, Brazil) has developed a cultivar, the BRS-Pará, suitable for growing on stable land, making the production more high-yielding than the traditional system [6].

When fully ripe, the açaí fruit has a deep purple peel (epicarp), a small fleshy pulp (pericarp), and a single seed covered by a thin mesocarpic fiber (endocarp). Only 15% (*w*/*w*) of açaí is edible, generating seeds as the primary agricultural residue from depulping [2,7,8,9]. A small part is used in jewelry craft, food for pigs, decomposed into potting soil for home gardens, or burned for fuel, but most of it is just thrown away after pulp production [3].

An increasing number of publications has reported that polyphenols such as catechin, epicatechin, and oligomeric proanthocyanidins (PACs) are extractable from açaí seeds [10,11,12]. Reports with medicinal applications for the açaí seed extracts also showed protective effects against emphysema, lung inflammation, oxidative stress, cardiovascular dysfunction, and many other biological activities [13,14,15]. Most of them correlated PACs in açaí seed extracts with the observed pharmacological effects.

PACs are widespread in the Plant Kingdom, being the second largest group of polyphenols after lignins. They are composed of polyphenolic oligomers or polymers consisting of units of flavan-3-ol linked mainly through C4 to C8 or C4 to C6 interflavan bonds (B-type), and even an additional ether bond O7 to C2 or O5 to C2 (A-type) [16].

Despite its pharmacological properties, the bioavailability of PACs is primarily influenced by their degree of polymerization (DP). Oligomers with DP ≥ 4 do not pass the gut barrier, thus having limited applicability as pharmaceuticals [17]. However, high-weighted oligomers have been explored as an abundant, inexpensive, green source for adhesives, coatings, and resin productions for the industry [18,19,20], which prevents their accumulation in nature [21].

Corrosion is a material deterioration caused by chemical, electrochemical, and biological reactions. The global cost of corrosion is estimated at USD 2.5 trillion [22]. Effective prevention practices could result in substantial savings of 15–35% of the annual corrosion cost damage [22]. Natural products have been widely tested for metal protection from corrosion. The exploitation of agro-industrial residues’ anticorrosive potential has risen as an eco-friendly alternative, especially when considering the high availability, low cost, and potential for anticorrosive compounds of biomass wastes that garner less environmental concern and have fewer harmful effects than synthetic anticorrosives [23]. PAC extracts inhibit corrosion in several ways: as metal surface treatment agents, as oxygen scavengers, or as films (coating and cathodic film former) [24]. Mangrove (*Rhizophora* spp.) and coconut (*Cocos nucifera* L.) extracts enriched with these compounds, for example, have shown corrosion inhibition properties in different pH solutions [25,26,27].

A crucial step towards the agreement with global priorities and assessments of the United Nations (UN) 2030 Sustainable Development Agenda is using renewable raw materials in the sustainable production of by-products [28]. In this respect, Brazil has excellent potential to exploit agro-industrial residues, such as açaí seeds. Preliminary studies showed that PACs are a large part of the seed’s composition. This fact led us to the investigation of whether seed extracts of different açaí varieties (purple, white, and BRS-Pará) would possibly work as a source of bioactive PACs, which could be exploited in pharmaceuticals, food, and cosmetic applications. Furthermore, the determination of their anticorrosive properties could be a cheap, abundant, and green corrosion inhibitor, adding value for this by-product of the açaí industry.

## 2. Results and Discussion

The potential health benefits attributed to açaí seed extracts [29,30,31] highlight how critical comprehensive chemical characterization is. Research on the seed extract’s new biological effects is associated with its composition, thus adding value to this residue. Therefore, we started to investigate the chemical composition of varieties of açaí seeds: purple açaí (PA), white açaí (WA), and BRS-Pará (BRS).

The results from the extraction and liquid–liquid partitioning yields are shown in Table 1. The extraction generated 6.99% (*w*/*w*), 7.61% (*w*/*w*), and 8.04% (*w*/*w*) yields for BRS, WA, and PA, respectively. Liquid–liquid partitioning with ethyl acetate and water 1:1 (*v*/*v*) can separate PACs by size since only small weighted ones and other polyphenols are soluble in ethyl acetate. The aqueous fractions yielded above 75%, indicating that all varieties exhibited a high number of polymeric PACs.

After extraction, the content of PACs was quantified by n-butanol/hydrochloric acid assay following a published protocol [32]. PA exhibited 22.4% (*w*/*w*) proanthocyanidins/dry matter, while WA displayed 6.4% (*w*/*w*) and BRS 11.5% (*w*/*w*). Thus, these results show that açaí seed can be exploited as a source of polymeric proanthocyanidins.

The crude extracts were also analyzed using hydrophilic interaction chromatography (HILIC). A diol column separates PACs by size, allowing the report of results to be given in degree of polymerization (DP) after calibration [33]. A fluorescence detector (FLD) and mass spectrometry detector (micro-TOF) were used for comparison. Commercially available standards of epicatechin and procyanidin B2 were injected for retention time calibration for both methods (data not shown). Using the HILIC–HPLC–FLD analysis (Supplementary Material, Appendix A), PA, WA, and BRS crude extracts displayed individual peaks representing a degree of polymerization from 1 up to 12. The asymmetry in the peaks could indicate different PAC types in the samples or low column resolution [34].

The extracted ion chromatograms (EIC) from the HPLC–DAD–ESI–TOF–MS analysis (Supplementary Material, Appendix A–S4) were obtained using the molecular formula for procyanidins. The retention time showed peaks eluting in the increasing order of DP. PA and BRS crude extracts exhibited *m/z* 289, 577, 865 and 1153 (Δ = 288) of [M–H]^–^ characteristic of (epi)catechin and B-type, dimer, trimer, and tetramer, respectively. The *m/z* 720, 864, and 1008 (Δ = 144) represented the [M–2H]^2–^ of B-type procyanidin pentamer, hexamer, and heptamer. The *m/z* ion of 575 was indicative of [M–4H]^4−^ B-type procyanidin octamer. WA crude extract exhibited the same profile except for a B-type procyanidin octamer signal.

The results obtained from HILIC–HPLC–FLD and HILIC–HPLC–DAD–ESI–TOF–MS are comparable. However, it also displays how the disparity between the results can arise, for instance, from analytical differences. Both methodologies showed that açaí seeds (*E. oleracea* Mart.) crude extracts are composed of oligomeric procyanidins, even though HILIC–HPLC–FLD indicated a higher degree of polymerization in the extracts.

The ethyl acetate fractions were analyzed by direct infusion ESI–MS/MS in negative mode (Table 2). PA and BRS displayed an *m/z* 289.2 ion, with MS^2^ fragments with *m/z* 245 (a CO_2_ loss) and *m/z* 205 (C_4_H_4_O_2_ loss from the A-ring) ions indicative of (epi)catechin. An *m/z* 421.3 (289 + 132 Da) ion characteristic of a (epi)catechin-pentoside with a fragment of *m/z* 289.34 was found. Both samples exhibited an *m/z* 577.2 ion with MS^2^ fragments of *m/z* 425 and *m/z* 407 (425-H_2_O) characteristic of a retro-Diels–Alder (RDA) fission of B-type procyanidin dimer. It also displayed a MS^2^ fragment of *m/z* 451, a product of heterocyclic ring fission (HRF), and MS^2^ fragments of *m/z* 287 and *m/z* 289, indicative of a quinone methide fission [35].

The WA EtOAc fraction exhibited the same ions and fragmentation pattern of PA and BRS varieties, except for an unidentified compound with *m/z* 469 ion and an MS^2^ of *m/z* 289. This confirms that procyanidins are the main polyphenols in the açaí seeds, as corroborated by the literature [10,11,12,36].

As established by the liquid–liquid partitioning, the aqueous fractions are rich in polymeric PACs. Therefore, it was used for phloroglucinolysis of all varieties, and the results are shown in Table 1 (chromatograms are available in Supplementary Material, Appendix A). All varieties exhibited similar results, with a subunit composition profile of (−)-epicatechin as the single extension subunit found and a high percentage for (+)-catechin as a terminal subunit (above 80%). Their mean degree of polymerization (mDP) was also comparable, PA exhibited an mDP of 10.29, WA had an mDP of 11.23, while BRS aqueous fraction displayed an mDP of 11.81. Conversion yields for all the samples were above 80% which favors the homogeneity of the results. All açaí varieties presented a high polymerized B-type procyanidin. A previous study indicated similar results for açaí seed extracts with a high (−)-epicatechin content for extension subunit (above 95%), a high (+)-catechin percentage as terminal subunit (above 60%), and an mDP between 9.7–13 [11], corroborating that açaí seed extract might be a source of bioactive procyanidins, adding value to this by-product while decreasing the environmental impact.

MALDI-TOF mass spectra of PA, WA, and BRS aqueous fractions (Supplementary Material, Appendix A) showed clear repetitive patterns of peaks that allow the identification of specific oligomer series. Table 3 shows the possible combinations of different repeating units and the corresponding degrees of polymerization. All peaks correspond to sodium ion adducts (+23 Da) in the positive ion mode. The PA aqueous fraction had peak ions, indicating the presence of a B-type procyanidin series of DP = 3 up to 11 (889.7, 1177.8, 1466.1, 1754.5, 2042.6, 2330.8, 2619.0, 2907.0, 3195.4) because of repeating units (Δ = +288 Da) characteristic to procyanidin units. Results also show a heterogeneous series with one prodelphinidin unit substitution (904.8, 1193.3, 1481.5, 1770.0, 2058.0, 2346.0, 2634.3, 2921.9, 3211.0) with DP = 3 to 11 and B-type linkages. A variation of *m/z* +16 units is indicative of the presence of a hydroxyl group. Furthermore, a B-type procyanidin series with one galloyl substitution was identified (1329.4, 1618.6, 1906.5, 2193.4, 2483.5, 2771.0, 3058.1) with DP = 4 up to 10 (galloyl esterification displays a Δ = +152 Da). The BRS aqueous fraction also had the same pattern with a B-type procyanidin series (DP = 3–11), a procyanidin–prodelphinidin heterogeneous series (DP = 3–11), and a B-type procyanidin with one galloyl substitution (DP = 4–10). WA showed similar patterns with a B-type procyanidin (with DP = 3–11) and heterogeneous procyanidin–prodelphinidin sequences (with DP = 3–10). However, it did not have a galloyl substitution sequence.

Heterogeneous sequences with procyanidin–prodelphinidin units in other plants are commonly reported in the literature [27,37]. All açaí varieties exhibited an arrangement with a one-unit exchange from procyanidin for prodelphinidin. Nevertheless, the pholoroglucinolysis reaction and MALDI-TOF data were consistent, showing that B-type procyanidins are the major components in the aqueous fractions of *Euterpe* spp. analyzed.

The *E. oleracea* Mart. seeds (PA, WA, and BRS varieties) are by-products with no primary exploration and with ecological implications due to accumulation. PA seed extract displayed a higher concentration of PACs (BuOH/HCl assay) and, therefore, was chosen for the corrosion experiments.

The PA crude extract corrosion inhibition activity was analyzed after 24 h of immersion in corrosive solution by fitting an LPR curve with ANOVA software to calculate the Tafel parameters, shown in Table 4, and by potentiodynamic curves (Figure 1). The PA crude extract only promoted an expressive effect at the concentration of 1.0 g/L, both increasing the corrosion potential (E_corr_) and polarization resistance. The effect on E_corr_ suggests an anodic type of corrosion inhibitor, but the increasing polarization resistance is typical for the cathodic type of corrosion inhibitor. Therefore, PA crude extract behavior suggests inhibition for both cathodic and anodic reactions [38,39].

The PA crude extract efficiency as a corrosion inhibitor was estimated by corrosion current density (J_corr_) and by polarization resistance (Rp) of the metallic surface to evaluate the metal susceptibility to corrosion and the stability of the protective film on the metal surface, respectively. Table 4 shows that the anticorrosion activity occurred only at the concentration of 1.0 of PA crude extract. This data was corroborated by corrosion rate, also shown in Table 4. The efficiency of a corrosion inhibitor depends on a concentration range with an ideal number of molecules to form a stable, protective film. The absence or abundance of molecules leads to unstable film formation, either due to an insufficient number of molecules or the dispute of metallic surface interaction by excess of molecules [39].

The corrosion inhibitor efficiency of over 99.9% and a considerable reduction of corrosion rate shows that 1.0 g/L of PA crude extract, in neutral pH after 24 h of immersion, is a promising green corrosion inhibitor. Furthermore, the efficiency obtained for 1.0 g·L^−1^ of PA crude extract is higher than studies with PACs from other natural sources [25,26].

The Tafel slopes of 0.1 to 0.8 g/L and corrosion rate data showed a more intense corrosive process than that observed for the control (absence of inhibitor). These results prove that a weak corrosion inhibitor can promote the corrosive process [39,40]. However, the significant reduction on the anodic (β_anodic_) and cathodic (β_cathodic_) coefficients in the presence of 1.0 g/L^1^ of PA crude extract shows the importance of the ideal concentration of inhibitor for inhibition of corrosion reactions.

The potentiodynamic curves (Figure 1) corroborated the suggestion that PA crude extracts affect anodic and cathodic reactions. After 24 h of immersion, in the condition of 1.0 g/L of PA crude extract, both anodic and cathodic branches showed low values of current density compared to the control. The other tested concentrations showed similar curves to those observed for the control.

Adsorption inhibitors affect the corrosion reaction in anodic and cathodic branches, as observed in the presence of 1.0 g/L of PA crude extract with a significant reduction of corrosion current density (Figure 1). This behavior is standard in organic compounds, such as PACs, for which there are already records in the literature [25,26,27,41]. The protective film was formed by adsorption of molecules on the metal surface, as the black/dark purple film on the metal surface in 1.0 g/L of PA crude extract was observed (Supplementary, Appendix A).

The effect of PACs on the metal surface was analyzed through EIS, displayed as Nyquist and Bode plots (Supplementary Material, Appendix A). The impedance consists of the relation between the alternating potential and the alternating current in the response of known frequency applications. Although the impedance is inversely proportional to the current, higher impedance modules promote lesser corrosion susceptibility. For the Nyquist plot in the presence of PA 1.0 g/L crude extract, the semicircle suggests the formation of a protective layer, blocking or reducing the effect of corroding molecules of the electrolyte on the metal surface. In the Bode plot with PA 1.0 g/L crude extract, there is a peak of the phase angles at low frequency, corroborating the protective film formation on the metallic surface hypothesis. Thus, as observed in the Nyquist plot, the presence of PACs inhibits the corrosion process. Furthermore, lower concentrations (0.2–0.8 g/L) probably do not have a sufficient number of molecules to form an efficient protective layer. However, in 1.0 g/L, the molecules content is ideal for metal surface adsorption, avoiding corrosive processes.

Previous studies have revealed açaí seeds as a promising source of antioxidants, with a higher antioxidant activity than the pulp bioactive compounds [7,11,12]. However, this is the first time that açaí seed extracts were tested for corrosion inhibition activity. In the PA crude extract, the high radical scavenging capacity and reducing metal activity of PACs translated to a promising green corrosion inhibitor for carbon steel AISI 1020 in neutral pH conditions at room temperature.

The protective activity can be affected by inhibitor chemical composition, metal type, and experimental physicochemical conditions. These variables can affect (positively or negatively) the interaction between the inhibitor and the metallic surface and consequently its anticorrosion activity [42,43,44]. Especially for PACs, their origin affects the polymerization degree, and it involves the interaction with the metal surface and the stability of the protective film. It was previously described that polymeric PACs are more effective for metal surface protection than monomeric ones [27,42,43,44].

Crude extracts from açaí (*E. oleracea* Mart.) seeds and coconut (*Cocos nucifera* L.) husk fiber are natural sources of phenolic compounds. Comparing the polymeric degree of PACs from these natural sources showed that those PACs from açaí seed crude extracts had a higher polymeric degree than those from coconut husk fiber [25]. Furthermore, the PA crude extract proved to be more effective (higher inhibition efficiency and lower concentration) for carbon steel metal protection than coconut husk fiber crude extract. This supports the relation between high polymerization degree and corrosion inhibition activity.

## 3. Materials and Methods

### 3.1. Materials

Analytical grade acetone and HPLC grade methanol, acetonitrile, ethyl acetate, acetic acid, and formic acid used were obtained from TEDIA (Rio de Janeiro, Brazil). Water was purified using a Milli-Q system from Millipore (Billerica, MA, USA). Phloroglucinol, (+)-catechin, (−)-epicatechin, and procyanidin B2 (dimer) were purchased from Sigma-Aldrich (St. Louis, MO, USA), and NaHCO_3_ was acquired from Spectrum (New Brunswick, NJ, USA).

### 3.2. Extraction Procedures

*E. oleracea* Mart. seeds from purple açaí (PA) and white açaí (WA) were obtained in São João de Pirabas (Pará, Brazil) by a local private producer. The *E. oleracea* Mart. BRS-Pará (BRS) seeds were graciously donated by Embrapa Oriental Amazonia (Belém, Pará). The seeds had their pulps removed and were washed, air-dried, and transported to the laboratory. The seeds were grounded, defatted (20.0 g, triplicate) by Soxhlet extraction with n-hexane (3 × 4 h). The defatted powdered seeds were extracted (2.0 g, triplicate) with Me_2_CO:H_2_O 6:4 in an ultrasound bath for three 10 min cycles, renewing solvent in each period. The crude extracts were vacuum filtered; the acetone was evaporated at 35 °C in a rotatory evaporator and then lyophilized.

### 3.3. Ethyl Acetate:Water Liquid–Liquid Partitioning

Crude extracts were dissolved in 50 mL of water and then partitioned with the same volume of ethyl acetate three times. The organic phase was evaporated in a rotatory evaporator, and the aqueous fraction was lyophilized. 

### 3.4. Proanthocyanidin Content by n-BuOH/HCl Test

The determination of proanthocyanidin content followed a published protocol [30]. The grounded and defatted seeds (0.2 g in triplicate) were submitted to ultrasound-assisted extraction with acetone:water 7:3 (10 mL) for 10 min. A 5 µL aliquot of extract was added to a sealed test tube containing 3 mL of n-butanol/hydrochloric acid (95:5) and 100 µL of a 2% solution of NH_4_Fe(SO_4_)_2_ in 2N hydrochloric acid. The solution was heated for 60 min (95 °C). After cooling, the absorbance was measured at 550 nm (UV-1601 Shimadzu spectrophotometer). Results were expressed in percentage (%) by dry matter.

### 3.5. HILIC–HPLC–FLD Proanthocyanidin Analysis

The HPLC (Perkin-Elmer Flexar) was equipped with a LC column oven, LC autosampler, quaternary LC pump, solvent manager, PDA, FLD detectors, and Chromera-Flexar software. A previously published protocol was followed [31], where samples were prepared by dissolving 1.0 mg of crude extract in acetonitrile/water 1:1 to 1.0 mg/mL and filtering through 0.22 µm PTFE filters. Aliquots of 10 µL were injected in a Merck LiChrospher^®^ 100 diol column (250 × 4.6 mm, 5 µm) with a LiChroCART guard column of the same material. The flow rate was 0.8 mL/min; the column temperature was 30 °C, the fluorescence detector was set with excitation at 276 nm and emission at 316 nm. A binary mobile phase was used, namely acetonitrile:acetic acid (98:2) as phase A, and methanol:water:acetic acid (95:3:2) as phase B. Gradient elution: 0–35 min, 0–40% B, 35–40 min, 40% B, 40–45 min, 40–0% B. The column was washed for 10 min with 90% B followed by 20 min as re-equilibration time.

### 3.6. MALDI-TOF-MS

The mass spectrometry analyses were made on a MALDI-TOF Autoflex Speed Bruker spectrometer using a published protocol [35] with slight modifications. Samples (2 mg) were dissolved with 0.1% aqueous trifluoroacetic acid and diluted to 1:10 with DHB matrix solution (0.1% TFA). A 1.0 µL aliquot of aqueous sodium chloride (1.0 mg/mL) was added to the solution. The samples were applied to the MALDI plate (duplicate) and analyzed after they were dried. Analyses were made in positive mode, using a peptide calibration standard (Bruker, Billerica, MA, USA). Data were processed using mMass 5.5.0 software (Strohalm M.^®^).

### 3.7. Direct Infusion ESI-MS/MS Analysis

Ethyl acetate fractions were analyzed by direct infusion in an Amazon SL ESI-iontrap spectrometer (Bruker). Samples (1.0 mg) were diluted 1:100 with methanol and analyzed in negative mode. Infusion flow rate: 3 µL/min, nebulizer pressure: 10 psi, N_2_: 5 L/min, source temperature: 200 °C.

### 3.8. Phloroglucinolysis

Phloroglucinolysis determinations used a previously published protocol [45] with a phloroglucinol solution prepared with 5.0 g of phloroglucinol, 1.0 g of ascorbic acid, and 0.1 N HCl in a minimum amount of methanol up to 100 mL. NaHCO_3_ solution was prepared with 336.0 mg of NaHCO_3_ and Milli-Q water up to 100 mL (40 mM).

Capped glass test tubes containing 5.0 mg weighted samples (triplicate) with 1.0 mL of the phloroglucinol solution were heated for 20 min in a water bath (50 °C). An aliquot of 200 µL was transferred to a 2.0 mL vial, and 1.0 mL of the NaHCO_3_ solution was added. A ReproSil-Pur RP-18 column (250 × 4.6 mm, 5 µm, Dr. Maisch GmbH, Germany) with a guard column of the same material was used; flow rate: 1 mL/min, 280 nm UV detector. Mobile phase (A) was aqueous 1% acetic acid, and (B) was methanol. Gradient mode: 5% B for 10 min, 5–20% B in 20 min, 20–40% B in 25 min, 90% B for 10 min, 5% B for 5 min.

### 3.9. HPLC–DAD–ESI-TOF-MS

Analysis followed a protocol [46] with modifications. The HPLC–DAD analysis occurred using·LiChrospher^®^ 100 diol (250 × 4.6 mm, 5 µm, Merck) with guard column LiChroCART of the same material at 30 °C. Two solvents were used: water with 1% (*v*/*v*) formic acid (A), and acetonitrile (B). The elution profile was 0–40 min, 95–35% (*v*/*v*) B in A (linear gradient); 40–45 min, 35–95% B in A (linear gradient); and 45–75 min 95% B in A (isocratic). The flow rate was 0.3 mL/min; λ = 280 nm. Injection volume was 10 μL (1 mg/mL). ESI-TOF-MS was in a Bruker’s MicrOTOF II in negative mode, scan from m/z 100 to 1500, capillary set at 3000 V, offset endplate at −500 V, nebulizer at 4.0 bar, dry heater at 200 °C, and dry gas at 10.0 L/min. Extracted ion chromatograms (EIC) were obtained using the molecular formula for procyanidins with ±0.01 accuracy and intensity from 10^5^ to 1000.

### 3.10. Corrosion Experiments

The corrosion tests were carried out in a three-electrode (a working electrode, a reference KCl-saturated calomel electrode (SCE), and a platinum counter electrode) electrochemical cell which was connected to an AUTOLAB PGSTAT302N potentiostat/galvanostat. Square coupons of carbon steel AISI 1020 (Fe–98.5%, C–0.2%, Mn–0.6% and traces of P, S, Si, Sn, Cu, Ni, Cr, and Mo) were used as working electrodes. The corrosive solution simulates cooling systems water and was composed of 500 ppm chloride (829 mg/L NaCl), 150 ppm of sulfate (222 mg/L Na_2_SO_4_), and 150 ppm of calcium carbonate (CaCO_3_). The pH electrolyte was controlled in neutral range (7.0 ± 0.2) during all immersion times with different concentrations of PA crude extract (0–1.0 g/L).

These tests were conducted in 800 mL glass cells, maintained at controlled temperature (25 ± 2 °C), and slow shaken for 24 h for data acquisition. Before each test, carbon steel AISI 1020 coupons (working electrodes), with 1.0 cm^2^ of the exposed area, were mechanically polished using SiC papers of 80, 120, 220, 320, 400, 500 and 600 grit, washed with distilled water, cleaned with ethanol and dried in hot air. And, the lyophilized PA crude extract was resuspended in distilled water and sterilized by membrane filtration (0.22 mm pore) to prepare a stock solution, used as basis for the tests PA crude extract solutions.

The activity as corrosion inhibitor was determined through the comparison between the results in absence and presence of different concentrations of PACs in PA crude extract and performed using LPR and potentiodynamic curves techniques. The concentrations tested were 0.1, 0.2, 0.5, 0.8 and 1.0 g/L. The potentiodynamic polarization curves were performed by working electrode scanning from −800 to −500 mVSCE at a scan rate of 0.33 mVs^−1^. And the Tafel data acquisition applied linear polarization resistance (LPR) technique by working electrode scanning from −20 to 20 mVSCE around the corrosion potential at a scan rate of 0.33 mVs^−1^. The inhibition efficiencies were calculated trough two data. The first method applied Tafel data (Equation (1)), where $J_{corr}$ and $J_{corr}^{0}$ are the corrosion current density values with and without inhibitor, respectively. The second one by linear polarization resistance (LPR) data (Equation (2)), where $R_{p}$ and $R_{p}^{0}$ are the resistance polarisation values in the presence and absence of inhibitor.
(1)$$\eta_{Tafel}\left(\%\right)=\frac{J_{corr}^{0}-J_{corr}}{J_{corr}^{0}}\;\times \;100$$
(2)$$\eta_{LPR}\left(\%\right)=\frac{R_{p}-R_{p}^{0}}{R_{p}}\;\times \;100$$

## 4. Conclusions

In this study, all analytical techniques determined that the three açaí varieties (PA, WA and BRS-Pará) are a potential source of PACs with a high degree of polymerization, which could have industrial applications. All varieties exhibited a similar chemical composition, indicating that the seeds do not need to be separated to exploit their phenolics. Furthermore, PA crude extract showed itself as a promising green corrosion inhibitor for carbon steel AI-SI 1020 in neutral pH corrosive solutions at room temperature. A black/dark purple film was observed on carbon steel AISI 1020 surface after 24 h of immersion in the presence of 1.0 g/L of PA crude extract, suggesting the inhibitor adsorption and a protective film formation. The inhibition efficiency of over 99.9% indicated the açaí seed extract is a promising target for future studies about corrosion inhibitors from natural sources, especially from a raw solid waste as the açaí seeds.

## Figures and Tables

**Figure 1 molecules-26-03433-f001:**
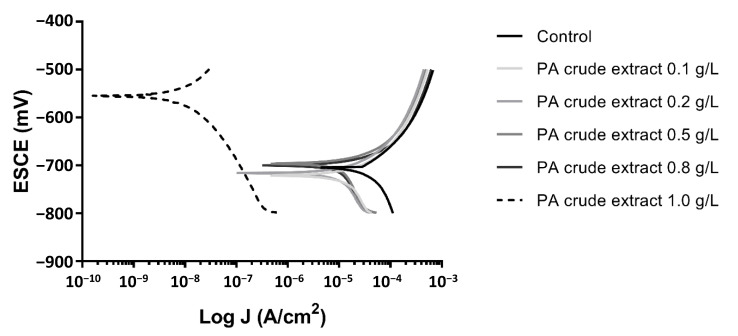
Potentiodynamic curves of carbon steel AISI 1020 after 24 h in neutral pH corrosive solution for different concentrations of PA crude extract.

**Table 1 molecules-26-03433-t001:** Extraction, liquid–liquid partition, mDP, and n-BuOH/HCl result from PA, WA, and BRS samples.

Results/Samples		PA	WA	BRS
Extraction yield (%)		8.04 (±0.85)	7.61 (±0.78)	6.99 (±1.25)
Liquid–Liquid Partitioning	EtOAc fraction yield (%)	9.6	15.6	8.7
	Aqueous fraction yield (%)	87.4	83.1	75.9
PAC content (%/dry matter)		22.4 ± 5.0	6.4 ± 0.6	11.5 ± 3.8
mDP by acid catalysis		10.29 ± 0.01 [0.07]	11.23 ± 0.09 [0.84]	11.81 ± 0.09 [0.74]
Terminal subunit composition in (%) of (+)-catechin		86.22 ± 0.19 [0.22]	83.28 ± 0.34 [0.41]	84.91 ± 0.51 [0.61]
Conversion yield (%)		98.7	84.3	121.5

PA—purple açaí, WA—white açaí, BRS—BRS-Pará açaí cultivar, EtOAc—ethyl acetate, mDP—mean degree of polymerization.

**Table 2 molecules-26-03433-t002:** PA, WA, and BRS EtOAc fractions by ESI–MS/MS direct infusion.

PA		WA		BRS		
Molecular ion [M − H]^−^ (*m/z*)	MS^2^ (*m/z*)	Molecular ion [M − H]^−^ (*m/z*)	MS^2^ (*m/z*)	Molecular ion [M − H]^−^ (*m/z*)	MS^2^ (*m/z*)	Identification
289.1	245; 205	289.1	245; 205	289.1	245; 205	(epi)catechin
577.2	425; 407; 289; 451; 287	577.2	407; 425; 289; 451; 287	577.2	451; 425; 407; 289; 287	B-type procyanidin dimer
421.28	289	421.2	289	421.25	289	(epi)catechin-pentoside
-	-	469.1	289	-		Unknown compound

PA—purple açaí, WA—white açaí, BRS—BRS-Pará açaí cultivar.

**Table 3 molecules-26-03433-t003:** PA, WA, and BRS aqueous fractions MALDI-TOF [M+Na]^+^ mass spectra.

		PA	WA	BRS			
DP	Predicted [M + Na]^+^ (Da)	Observed [M + Na]^+^ (Da)	Observed [M + Na]^+^ (Da)	Observed [M + Na]^+^ (Da)	Monomeric Units		
					(epi)catechin	(epi)gallocatechin	Galloyl
3	889.8	889.7	889.9	889.6	3	0	0
	905.8	904.8	905.8	905.5	2	1	0
4	1178.0	1177.8	1178.2	1177.9	4	0	0
	1194.0	1193.3	1194.2	1193.8	3	1	0
	1330.1	1329.4	-	1330.0	4	0	1
5	1466.3	1466.1	1466.6	1466.2	5	0	0
	1482.3	1481.5	1482.8	1482.0	4	1	0
	1618.4	1618.6	-	1618.3	5	0	1
6	1754.5	1754.5	1755.7	1754.4	6	0	0
	1770.5	1770.0	1771.3	1770.1	5	1	0
	1906.6	1906.5	-	1906.4	6	0	1
7	2042.8	2042.6	2043.9	2042.7	7	0	0
	2058.8	2058.0	2060.1	2058.6	6	1	0
	2194.9	2193.4	-	2194.4	7	0	1
8	2331.0	2330.8	2332.6	2330.9	8	0	0
	2347.0	2346.0	2347.8	2346.3	7	1	0
	2483.1	2483.5	-	2483.5	8	0	1
9	2619.3	2619.0	2621.3	2619.1	9	0	0
	2635.3	2634.3	2636.4	2634.4	8	1	0
	2771.4	2771.0	-	2770.9	9	0	1
10	2907.6	2907.0	2907.5	2907.1	10	0	0
	2923.5	2921.9	2924.6	2922.6	9	1	0
	3059.6	3058.1	-	3059.7	10	0	1
11	3195.8	3195.4	3196.2	3195.1	11	0	0
	3211.8	3211.0	-	3210.6	10	1	0

**Table 4 molecules-26-03433-t004:** Galvanostatic and linear polarization resistance (LPR) parameters for AISI 1020 carbon steel in neutral pH with various PA concentrations after 24 h of immersion.

Concentration (g/L)	Tafel Data					LPR Data		Corrosion Rate (mm/year)
	E_corr_ (mV, SCE)	β_anodic_ (mVdec^−1^)	β_cathodic_ (mVdec^−1^)	J_corr_ (mAcm^−2^)	η_Tafel_	Rp (Ωcm^2^)	η_LPR_	
0	−732	0.03819	0.03836	6.215 × 10^−6^	-	1.22 × 10^3^	-	0.072216
0.1	−696	0.05404	0.14260	2.465 × 10^−5^	0	633.6	0	0.28638
0.2	−701	0.06467	0.02803	2.298 × 10^−5^	0	724.9	0	0.26704
0.5	−698	0.03672	0.10310	1.254 × 10^−5^	0	710.6	0	0.14569
0.8	−700	0.04628	1.74730	2.334 × 10^−5^	0	749	0	0.27126
1.0	−556	0.00655	0.00592	5.122 × 10^−10^	99.99	1.75 × 10^6^	99.93	5.92 × 10^−6^

Ecorr—corrosion potential; Jcorr—corrosion current density; ηTafel—inhibition efficiency calculated with Jcorr values and zero standardized for negative values; Rp—polarization resistance; ηLPR—inhibition efficiency calculated with Rp values.

## Data Availability

Not applicable.

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
