# Peer review of "Açaí (*Euterpe oleracea* Mart.) Seed Extracts from Different Varieties: A Source of Proanthocyanidins and Eco-Friendly Corrosion Inhibition Activity"

_molecules, 2021, doi:10.3390/molecules26113433_

Round 1
Reviewer 1 Report
- line 185; What is the corrosive solution? Please show the composition of test solution! Was the test solution deaerated or not?
- line 191; I don't agree the mixed-type corrosion inhibitor. In Figure 1, the slopes of the curves were not almost changed. So, this shows the different mechanism. Re-analyse them.
- Table 3; Ecorr, mV(SCE). Tafel slopes, Jcorr, LPR etc. had a significant error, and they must be calculated again.
- line 331; Corrosion test methods must be rewritten, including test condition, solution etc.
- line 349; Icorr etc. including eq. (1)(2) shall be rewritten.
- Why didn't you try the concentration over 1.0 g/L?
- Why didn't you try the surface analysis on the corrosion inhibited surface? Through the surface analysis, you can find the mechanism of eco-friendly inhibitor.
Reviewer 2 Report
COMMENTS ON MANUSCRIPT molecules-1225036
TITLE: Açaí (Euterpe oleracea Mart.) seed extracts from different varie-2 ties: a source of proanthocyanidins and eco-friendly corrosion 3 inhibition activity
COMMENTS
The article reports on the extraction and investigation the chemical composition of varieties of açaí seeds: 85 purple açaí (PA), white açaí (WA) and BRS-Pará (BRS) as well as evaluation of crude PA as a corrosion inhibitor for AISI 1020 carbon steel in neutral pH solution. The corrosion inhibition effect was investigated using electrochemical (LPR and PDP) techniques. The major finding of the work is the discovery of proanthocyanidins as the major constituent of varieties of açaí seeds. Although the authors have done an impressive work on the elucidation of the chemical composition of the varieties of açaí seeds using an array of analytical techniques, the same cannot be said of corrosion inhibition performance assessment of the extract of PA. Overall assessment shows that the work is well written and conclusions drawn are supported by the data. However, the corrosion inhibition studies aspect of this manuscript is very weak and the authors need to carry out a major revision before the manuscript can be considered for publication in Molecules. The issues to be addressed by the authors are appended below:
(1) The experimental procedures for the electrochemical techniques used (LPR and PDP) should be sufficiently described so that it can be reproduced by researchers working in the same area. For instance, the potential range in both PDP and LPR techniques as well as the scan rate for LPR is missing. They should be given.
(2) This statement in line 345 "The corrosion potential (Ecorr), current density (Jcorr), and corrosion rates were performed" is not correct and should be modified to read “The corrosion experiments were performed using LPR and PDP techniques". Corrosion potential (Ecorr), current density (Jcorr) and corrosion rates are electrochemical parameters derived from the techniques.
(3) The electrochemical parameters as well as corrosion inhibition efficiency for other extract concentrations (0.1-0.8 g) should be calculated and listed in Table 4.
(4) The authors need to add one more electrochemical technique-electrochemical impedance spectroscopy (EIS) to the work. This is important for comparison purposes with other electrochemical techniques used.
(5) Surface morphological characterization of the corroded steel samples without and with the crude PA extract (1.0 g/L) as inhibitor with SEM/EDX, AFM or XPS should be added to the manuscript. This is needed to confirm that the observed corrosion inhibition is attributed to adsorption of the major phytoconstituent of the PA extract on the steel surface.
Reviewer 3 Report
The manuscript entitled “Açaí (Euterpe oleracea Mart.) seed extracts from different varieties a source of proanthocyanidins and eco-friendly corrosion inhibition activity”, authored by Gabriel Rocha Martins and colleagues, deals with the investigation of the phytochemical profile of E. oleracea Mart. seeds from different varieties. The work is well written and interesting, however some clarifications and modifications must be made before it is considered acceptable as a publication on Molecules.
GENERAL CONSIDERATIONS:
I think the work is very interesting, but a little ambiguous. Several times, the authors stated the need for a chemical characterization, but they exclusively profiled and quantified a single class of metabolites. The authors may support their decision highlighting that they focused their attention on PACs because the chemical analysis via LC-MS/MS suggested that PACs were the major metabolites in seeds. However, this point should be better clarified in the abstract.
The term ‘proanthocyanidin’ is repeated several times in the manuscript. Authors should consider replacing it with the acronym PACs after the first appearance.
ABSTRACT:
- Line 20: with the sentence “their accumulation represents an environmental problem”, did the authors means that the seeds derived from food-processing of acai may be a potential source of pollution? Please, rewrite this sentence.
- Line 21: Please, modify “this work aimed to quantify and determine the phytochemical composition”.
KEYWORDS:
Keywords should be words not contained in the title, at most present in the abstract. Their usefulness is to make easier the searching of the article using the common scientific search engines. Since several keywords are already present in the title, and/or repeated several times in the abstract, I strongly advise the authors to change some of the proposed keywords with other news. As author guidelines clearly report, authors can provide up to 10 different keywords.
INTRODUCTION:
Generally, the introduction is well written. However, I believe that three points need to be further developed:
- the phytochemical characterization of acai fruit (pulp, peel and seed) should be better described. The Acai fruit is well known to contain several bioactive compounds that should at least mentioned.
- since the seeds of the fruit are actually considered as a food processing waste, the authors should point out that their research is aimed to valorise the potential bioactivity of a waste product. This valorisation may follow not only in a reduction of a environmental contamination but also in the possibility to formulate a new product with useful applications. In the recent years, other previously published work aimed more or less the same purpose were published. In particular, agronomic (root, leaves, etc…) and food processing (fruit peel, seeds, etc…) wastes were largely investigated for the extraction of bioactive compounds and for the formulation of both dietary supplements (org/10.3390/molecules25112612; doi.org/10.1016/j.jksus.2019.05.008; doi.org/10.3390/molecules24061032; 10.1007/s13205-020-02525-6); plant biostimulants (doi.org/10.3389/fpls.2020.00836; doi.org/10.3390/su13052710; 10.1038/s41598-020-79770-5), and other multiple biobased value-added products (doi.org/10.1021/acssuschemeng.9b07479).
- The potential application of natural extracts as anticorrosive agents is a not very well known application, but it could be really interesting. Since it could be a strength point of their manuscript, I suggest to authors to better describe this application in the introduction.
RESULTS: this section should be renamed as “Results and Discussion”.
- Table 1: authors should change the orientation of this table. Indeed, this would be much clearer if the species were in columns rather than rows. Furthermore, authors should report the meaning of each acronym as footnotes of the table.
- Table 2: authors should report the meaning of each acronym as footnotes of the table. Moreover, the term ‘Procyanidin monomer’ is incorrect. In reality, they are flavan-3-ols (catechins, epicathechins, etc.). Authors should fix this problem, both in this table and in the main text.
- Table 3: authors should report the meaning of each acronym as footnotes of the table.
- Table 4: authors should report the meaning of each acronym as footnotes of the table. Moreover, some number should be placed as apex (i.e. 1.254E-05).
- The anti-corrosive effect of the extracts evaluated by the authors could be related to the particular antioxidant properties of the extract. In fact, PACs are well known to be bioactive compounds with high radical scavenging capacity and reducing metal activity. Some simple spectrophotometric assays evaluate these properties. For example ABTS and DPPH effectively evaluate the radical scavenging properties, while FRAP, CuPRAC or similar assays evaluate the metal reducing properties. The authors could perform some of these essays, or at least recall in the discussion these properties evaluated on other types of extracts containing PACs .
MATERIALS AND METHODS:
- The equations should be reported as equations, using the Microsoft Word tool, and not as figure. Please fix the format.
- Also some mathematic elemen (i.e. Jcorr and J0corr) should be fixed in the main text.
- The acid butanol assay is a slightly dated method for evaluating the proanthocyanidin content. Other more reliable spectrophotometric methods (such as the DMAC-assay) have been developed and are currently considered to be more reliable. The authors, if they do not want to perform another essay, should better specify and motivate in the main text the reason why they have chosen this essay (10.3390/molecules25112612).
Round 2
Reviewer 1 Report
Line 223; What is the definition of mixed type inhibitor? You did already describe the formation of a protective film in line 226. Would you please re-check the corrosion mechanism about the inhibitor's type?
Author Response
Reviewer #1: Line 223; What is the definition of mixed type inhibitor? You did already describe the formation of a protective film in line 226. Would you please re-check the corrosion mechanism about the inhibitor's type?
Response:
Thank you for your considerations, we re-check the data and amended the manuscript accordingly. Our group considers the definition proposed by Sastri (2011) for mixed-type inhibitors.
[…] Mixed-type inhibitors affect both anodic and cathodic branches of a polarization curve. Organic compounds function as mixed-type inhibitors. The organic inhibitors adsorbed on the metal surface provide a barrier to dissolution at the anode and a barrier to oxygen reduction at the cathodic sites […].
So, we understand that adsorption inhibitors could be classified as anodic inhibitors (when inhibits the metal dissolution), cathodic inhibitors (when inhibits the oxygen reduction), or both/mixed-type inhibitors (when a protective film is formed and both anodic and cathodic reactions are inhibited). The manuscript was produced considering adsorption inhibitors and mixed-type inhibitors as synonyms. Still, after your considerations, we prefer to use only the adsorption inhibitor definition to corroborate the formation of a protective film on the metal surface.
The modifications can be found in the manuscript:
Page 7, lines 200-205: “The PA crude extract only promoted an expressive effect at the concentration of 1.0 g.L-1, both increasing the corrosion potential (Ecorr) and polarization resistance. The effect on Ecorr suggests an anodic type of corrosion inhibitor, but the increasing polarization resistance is typical for the cathodic type of corrosion inhibitor. Therefore, PA crude extract behavior suggests inhibition for both cahodic and anodic reactions [38,39].”
Page 7, lines 230-233: “The potentiodynamic curves (Figure 1) corroborated the suggestion that PA crude extract affect anodic and cathodic reactions. After 24 hours of immersion, in the condition of 1.0 g.L-1 of PA crude extract, both anodic and cathodic branches showed low values of current density compared to the control.”
Page 8, lines 239-241: “Adsorpiton inhibitors affect the corrosion reaction in anodic and cathodic branches, as observed in the presence of 1.0 g.L-1 of PA crude extract with a significant reduction of corrosion current density (Figure 1).”
Reference:
Sastri, V.S. Green Corrosion Inhibitors; John Wiley & Sons, Inc.: Hoboken, NJ, USA, 2011; ISBN 9781118015438. p. 107-108.
Reviewer 2 Report
The manuscript has been satisfactorily revised. It is recommended for acceptance in its current form.
Author Response
Reviewer#2: The manuscript has been satisfactorily revised. It is recommended for acceptance in its current form.
Response: We appreciate your helpful and valuable comments, as well as your favorable recommendation.
Reviewer 3 Report
After the revision, i think that the manuscript can be considered now as potential publication on Molecules
Author Response
Reviewer#3: After the revision, i think that the manuscript can be considered now as potential publication on Molecules
Response: We appreciate your helpful and valuable comments, as well as your favorable recommendation.